# Common Variation in Cytoskeletal Genes Is Associated with Conotruncal Heart Defects

**DOI:** 10.3390/genes12050655

**Published:** 2021-04-27

**Authors:** Fadi I. Musfee, A. J. Agopian, Elizabeth Goldmuntz, Hakon Hakonarson, Bernice E. Morrow, Deanne M. Taylor, Martin Tristani-Firouzi, W. Scott Watkins, Mark Yandell, Laura E. Mitchell

**Affiliations:** 1Department of Epidemiology, Human Genetics and Environmental Sciences, UTHealth School of Public Health, Houston, TX 77030, USA; Fadi.i.musfee@uth.tmc.edu (F.I.M.); a.j.agopian@uth.tmc.edu (A.J.A.); 2Department of Pediatrics, Perelman School of Medicine at University of Pennsylvania, Philadelphia, PA 19104, USA; goldmuntz@chop.edu (E.G.); hakonarson@chop.edu (H.H.); taylordm@chop.edu (D.M.T.); 3Center for Applied Genomics, The Children’s Hospital of Philadelphia, Philadelphia, PA 19104, USA; 4Department of Genetics, Albert Einstein College of Medicine, Bronx, NY 10461, USA; bernice.morrow@einsteinmed.org; 5Department of Biomedical and Health Informatics, The Children’s Hospital of Philadelphia, Philadelphia, PA 19104, USA; 6Division of Pediatric Cardiology, University of Utah School of Medicine, Salt Lake City, UT 84113, USA; martin.tristani@utah.edu; 7Nora Eccles Harrison Cardiovascular Research and Training Institute, University of Utah, Salt Lake City, UT 84112, USA; 8Department of Human Genetics, University of Utah, Salt Lake City, UT 84112, USA; swatkins@genetics.utah.edu (W.S.W.); myandell@utah.edu (M.Y.); 9Utah Center for Genetic Discovery, University of Utah, Salt Lake City, UT 84112, USA

**Keywords:** association, case-control, case-parent trios, congenital, conotruncal, heart, gene, genetics, genome-wide, malformation

## Abstract

There is strong evidence for a genetic contribution to non-syndromic congenital heart defects (CHDs). However, exome- and genome-wide studies conducted at the variant and gene-level have identified few genome-wide significant CHD-related genes. Gene-set analyses are a useful complement to such studies and candidate gene-set analyses of rare variants have provided insight into the genetics of CHDs. However, similar analyses have not been conducted using data on common genetic variants. Consequently, we conducted common variant analyses of 15 CHD candidate gene-sets, using data from two common types of CHDs: conotruncal heart defects (1431 cases) and left ventricular outflow tract defects (509 cases). After Bonferroni correction for evaluation of multiple gene-sets, the cytoskeletal gene-set was significantly associated with conotruncal heart defects (β_S_ = 0.09; 95% confidence interval (CI) 0.03–0.15). This association was stronger when analyses were restricted to the sub-set of cytoskeletal genes that have been observed to harbor rare damaging genotypes in at least two CHD cases (β_S_ = 0.32, 95% CI 0.08–0.56). These findings add to the evidence linking cytoskeletal genes to CHDs and suggest that, for cytoskeletal genes, common variation may contribute to the risk of CHDs.

## 1. Introduction

Congenital heart defects (CHDs) are the most common type of birth defect and are associated with significant morbidity and mortality [1,2,3]. Some individuals with CHDs have an identified malformation syndrome (e.g., 22q11.2 deletion, Holt-Oram). However, the majority of individuals with a CHD appear to be non-syndromic [4,5]. While a genetic contribution to non-syndromic CHDs has long been suspected, the number of genes for which there is strong evidence of an association with non-syndromic forms of human CHDs is relatively small [5,6,7].

A few CHD genes have been identified by classic linkage and candidate gene association studies [4]. In addition, single nucleotide polymorphism (SNP)-level, genome-wide association studies (GWAS) of CHDs and common variants have identified several genome-wide significant associations, although few of these associations have been independently replicated [8]. Further, studies using next generation sequencing data have revealed enrichment of potentially damaging de novo and rare inherited variants in specific gene classes. However, genome-wide significant associations with rare variants have been identified for only a few individual genes. For example, analysis of whole exome sequence (WES) data from 2871 CHD case-parent trios identified only seven genes with an excess of rare, potentially damaging de novo and inherited variants that exceeded genome-wide significance [9].

It is widely recognized that studies requiring genome-wide multiple testing corrections will identify only a small fraction of the disease-relevant genetic variation [10]. Consequently, analytic strategies that complement genome- and exome-wide analyses are used to extract additional information from such data. Gene-set analyses, which aggregate genetic variants and genes into biologically meaningful groups (e.g., based on function, expression pattern, biological pathway), and evaluate the overall effect of the set, provide one such strategy.

Gene-set analyses of de novo and rare inherited variants have provided additional insights regarding the genetic architecture of CHDs. For example, Watkins et al. [11] evaluated 15 CHD candidate gene-sets using WES data from 2391 trios and identified several sets (e.g., cilia, chromatin, and cytoskeletal genes) that were significantly enriched for rare damaging genotypes. Further, these analyses revealed differences in the variant profiles across gene-sets. For example, cilia genes were found to be enriched for rare inherited genotypes (recessive or compound heterozygous) and relatively depleted for rare de novo mutations, whereas the opposite pattern was observed for chromatin genes, and cytoskeletal genes were found to be enriched for both rare inherited and de novo genotypes.

Despite the insights into the genetic contribution to CHDs provided by gene-set analyses of rare variants, similar analyses have not been conducted using data on common variants. Gene-set analyses of common variants would, however, have increased power to detect associations relative to SNP- or gene-level GWAS. In addition to identifying new gene-sets that may be related to CHDs, such analyses would help to determine whether common and rare variants act through shared or distinct gene-sets. Consequently, to gain further insight into the genetic landscape of CHDs, we conducted common variant gene-set analyses for the 15 CHD candidate gene-sets that were assessed for rare genotypes by Watkins et al. [11]. Since CHDs are a heterogeneous group of conditions that may have overlapping but not identical risk profiles, we conducted our analyses separately for the two most common types of CHDs: conotruncal heart defects (CTDs) and left ventricular outflow tract defects (LVOTDs), as well as for the two defects in combination.

## 2. Materials & Methods

### 2.1. Data Sets

Our analyses were based on summary statistics from SNP-level genome-wide association meta-analyses of CTDs and LVOTDs, which, collectively, account for approximately 60% of all CHDs [12]. Details of the data and analyses underlying the summary statistics are published [13]. Briefly, meta-analyses were based on five datasets derived from study populations recruited through the Children’s Hospital of Philadelphia (CHOP) and the Pediatric Cardiac Genomics Consortium (PCGC). Informed consent was obtained from each case or the case’s parent/guardian, under protocols approved by the institutional review boards at CHOP or the PCGC clinical study sites. Individuals of all races and ethnicities were eligible to participate.

Cases with CTDs included individuals with tetralogy of Fallot, D-transposition of the great arteries, ventricular septal defects (conoventricular, posterior malalignment and conoseptal hypoplasia), double outlet right ventricle, isolated aortic arch anomalies, truncus arteriosus, or interrupted aortic arch. Cases with LVOTDs included individuals with hypoplastic left heart syndrome, coarctation of the aorta with or without bicuspid aortic valve, and aortic valve stenosis, and excluded individuals with variants of hypoplastic left heart syndrome, such as mal-aligned atrioventricular canal defects or double outlet right ventricle with mitral valve atresia. Medical records were reviewed to ensure accuracy of the cardiac phenotype. Cases with a known or suspected genetic syndrome (e.g., 22q11.2 deletion syndrome) were excluded.

Genotype data were generated using Illumina arrays followed by imputation. Prior to imputation, haplotypes were pre-phased using SHAPEIT2 v2.727 [14]. Genotype imputation was performed using Impute 2 v2.3.0 [15], with pre-phased data from the 1000 Genomes Project (Phase I integrated v3 variant set) as a reference. The following exclusions were made prior to imputation: case-parent trios with a Mendelian error rate > 1%; suspected duplicate samples (i.e., samples with pair-wise identity by descent > 0.6); and SNPs with a minor allele frequency (MAF) < 1%, genotyping rate < 90%, or deviation from Hardy Weinberg equilibrium in controls/parents (*p* ≤ 1 × 10^−5^). Post-imputation, individuals with genotyping rates < 90% were excluded, as were variants with poor imputation quality (r^2^ < 0.8), MAF < 5%, or genotyping rate < 90%.

SNP-level (MAF ≥ 0.05) GWASs, conducted in three CTD datasets: CHOP CTD trios (*N* = 670 trios), CHOP CTD cases/controls (*N* = 406 cases/2976 controls), and PCGC CTD trios (*N* = 355 trios), and two LVOTD datasets: CHOP LVOTD trios (*N* = 317 trios) and PCGC LVOTD trios (*N* = 192 trios), have been reported [13]. Briefly, the trio data were analyzed using a multinomial likelihood approach [16]. Genotypes were indexed using an additive (one degree of freedom) model of inheritance and, for each SNP, a likelihood ratio test was use to compare models with and without the genotypic parameter. These analyses were implemented in EMIM [17]. As these family-based analyses are robust to population stratification bias [18], cases of any race and ethnicity were included. The case-control data were analyzed in a similar manner using logistic regression as implemented in Golden Helix v8.1 (Golden Helix, Inc., Bozeman, MT, USA). The case-control analyses were restricted to include only Caucasian cases and controls and the SNP-phenotype associations were adjusted for the first two principal components of race/ethnicity. An additive genetic risk model was used in all analyses.

Meta-analyses, based on the summary statistics from the EMIM analyses of the individual studies, have also been reported [13]. Specifically, we conducted three meta-analyses: CTDs only (3 studies, 1431 cases); LVOTDs only (2 studies, 509 cases); and CTDs and LVOTDs combined (5 studies, 1940 cases). Meta-analyses were conducted using GWAMA v2.1 [19] with a fixed-effects model, unless there was evidence of heterogeneity (based on Cochran’s heterogeneity *p* ≤ 0.1), in which case a random-effects model was used.

### 2.2. Gene-Sets

We evaluated 15, previously-defined, CHD-related gene-sets [11]. These gene-sets were selected based on evidence of a role in heart development, or association with CHDs in humans or animals, and include sets based on gene function (e.g., cilia, cytoskeletal), pathways (e.g., hedgehog signaling), and expression patterns (e.g., mouse embryonic heart) (Table 1). The cytoskeletal gene-set excluded genes specific to cilia function and structure.

### 2.3. Gene-Set Analyses

Gene-set analyses were conducted using the regression-based approach implemented in MAGMA version 1.08 [20] and SNP-level summary statistics from our prior meta-analyses as input. Separate analyses were conducted for CTDs only, LVOTDs only, and for CTDs and LVOTDs combined. For these analyses, genes were defined by their transcription start-stop coordinates, based on the Genome Reference Consortium Human genome build 37. For each gene, we specified an annotation window that included 1 kilobase up- and downstream of these start-stop coordinates, and all SNPs located in the window were mapped to the gene.

Gene *p*-values were calculated from the summary statistics for SNPs within the gene annotation window. In MAGMA, gene *p*-values can be estimated by the mean of these statistics, the top statistic, or these two gene-level statistics can be combined into an aggregate statistic. We used the aggregate statistic, because it provides a more even distribution of power and sensitivity for a wider range of genetic models than the other gene-level statistics. The aggregate statistics were transformed to Z-scores using the probit function, such that associations with lower *p*-values are associated with higher Z-scores. The resulting Z-scores were used as the input for the gene-set analyses.

We used MAGMA to conduct competitive gene-set analyses using linear regression. The dependent variable in these analyses was the gene Z-score and the primary independent variable was a binary variable (S) indicating whether a gene is (S = 1), or not (S = 0) in the gene-set. Additional covariates were included, using the default options in MAGMA, to control for gene size, mean minor allele count in the gene and within-gene linkage disequilibrium. To account for linkage disequilibrium between genes in close proximity, MAGMA models the residuals as a multivariate normal, with correlations set to the gene-gene correlations estimated as part of the gene-level analyses. In MAGMA, gene–gene correlations are estimated for pairs of genes within five megabases of each other and are otherwise set to zero [20]. The gene-set statistic tests the null hypothesis that the mean association of the outcome (e.g., CTDs) with the genes in the set is greater than that of genes that are not in the set (i.e., H_0_: β_S_ = 0 versus H_1_: β_S_ > 0). A Bonferroni correction was used to account for the assessment of 15 gene-sets, such that gene-sets with *p* < 0.003 were considered to be significantly associated with the outcome.

For each significantly associated gene-set, we re-examined the results for each gene and SNP in the set, to determine whether these associations would be significant using a gene-set specific Bonferroni correction (i.e., *p* < 0.05/number of genes in the set). We also annotated each SNP in the gene-set with its odds ratio and 95% confidence interval from our prior SNP-level genome-wide meta-analyses [13], and with its location (e.g., intergenic) or consequence (e.g., missense mutation), and scaled a combined annotation dependent depletion (CADD) score [21,22] obtained using the Ensembl Variant Effect Predictor [23].

In addition, because the present analyses and the analyses of Watkins et al. [11] both used data from the PCGC, for each significant (*p* < 0.003) CHD-gene-set association, we determined the number of cases with potentially damaging rare genotypes in relevant genes (as identified by Watkins et al. [11]) in the analysis. These cases were not omitted from the current analyses, because we used SNP-level summary statistics from GWAS that pre-dated the work of Watkins et al. [11]. The number of cases that would have been excluded was, however, relatively small. For example, only 2% of the CTD cases in our analyses were both included in the analyses of Watkins et al. [11], and found to carry a potentially damaging rare genotype in a cytoskeletal gene.

### 2.4. Post Hoc Analyses

Although not part of our original analysis plan, we conducted additional analyses for each significant (*p* < 0.003) CHD-gene-set association, to assess whether the association was stronger when the gene-set was restricted to include only those genes that were found to harbor damaging de novo or rare recessive or compound heterozygous genotypes in at least one of the 2391 whole-exome sequenced CHD trios included in the analyses of Watkins et al. [11]. Specifically, we conducted competitive analyses for three restricted gene-sets including genes with: damaging de novo mutations in at least one CHD case; damaging recessive or compound heterozygous genotypes in at least one case; or damaging de novo mutations and/or recessive or compound heterozygous genotypes in more than one case. The magnitude of the association between the outcome and each of these gene-sets was compared using the gene-set indicator parameter estimates (i.e., β_S_).

## 3. Results

We used a regression-based approach to assess associations between 15 CHD-related gene-sets and the two most common types of CHDs (i.e., CTDs and LVOTDs). Our analyses were based on summary statistics from prior genome-wide, common variant (i.e., MAF > 5%) analyses of CTDs only, LVOTDs only and the combined CTD and LVOTD groups [13]. For each outcome (CTDs only, LVOTDs only, CTDs and LVOTDs combined), gene-level *p*-values were generated for 17,343 genes. Using a genome-wide Bonferroni correction (i.e., *p* < 0.05/17,343) no gene was significantly associated with any of these outcomes (Appendix A). Further, no gene-set was significantly associated (i.e., *p* < 0.003) with LVOTDs only, or with CTDs and LVOTDs combined. However, a significant association was identified between the cytoskeletal gene-set and CTDs (*p* = 0.001) (Table 2). The coefficient for the gene-set variable (i.e., β_S_ = 0.09) indicates that, on average, the Z-scores for genes in the cytoskeletal set are higher than the Z-scores for genes that are not in this set after controlling for gene size, mean minor allele count in the gene, and both within and between gene linkage disequilibrium, since these Z-scores are inversely related to their corresponding *p*-values.

We re-examined the gene-level *p*-values for genes in the cytoskeletal gene-set (Appendix A), using a Bonferroni correction for the number genes in the set (*N* = 726, *p* < 6.89 × 10^−5^). No gene was significantly associated with CTDs after this correction. We also re-examined the associations for SNPs mapping to genes in the cytoskeletal set (*N* = 131,628 SNPs). The odds ratios for the associations between these SNPs and CTDs ranged from 0.7–1.5 (Appendix A). Using a Bonferroni correction for the number of SNPs in the cytoskeletal set (*p* < 3.8 × 10^−7^), none of these SNPs were significantly associated with CTDs.

The 10 cytoskeletal genes with the lowest *p*-values are provided in Table 3. The lowest gene association *p*-value was for Cas scaffold protein family member 4 (*CASS4, p* = 0.0032), a cytoplasmic adaptor protein involved in integrin signaling pathways that are important for cell migration and adhesion [24]. Of the SNPs mapping to *CASS4*, the lowest *p*-value was *p* = 0.0002 for an intronic variant (rs2064860) with a scaled CADD score of 1.06. This gene also includes a variant in the 5′ untranslated region with a *p*-value < 0.05 and a CADD score ≥ 10 (rs17462136, *p* = 0.038, CADD = 18.66) (Appendix A).

For SNPs mapping to a cytoskeletal gene, the lowest (albeit non-significant) *p*-value was *p* = 6.6 × 10^−6^, for an intronic variant (rs12072230), with a scaled CADD score of 2.14, in kazrin periplakin interaction protein (*KAZN*), a cytoplasmic adaptor that binds to p120 catenin family members, which are important in maintaining cell shape integrity via the actin cytoskeleton [25]. An additional seven SNPs mapping to this gene had *p* < 0.05 and CADD ≥ 10, including four intronic variants (rs41269409, *p* = 0.008, CADD = 11.13; rs1721829, *p* = 0.028, CADD = 10.40; rs804127, *p* = 0.032, CADD = 13.07; rs761191, *p* = 0.045, CADD = 13.74), two intergenic variants (rs17399514, *p* = 0.011, CADD = 10.47; rs2697976, *p* = 0.016, CADD = 10.06;) and one variant in a regulatory region (rs10927460, *p* = 0.010, CADD = 11.76) (Appendix A). *KAZN* had the fourth lowest gene-level *p*-value (*p* = 0.007) in the cytoskeletal gene-set (Table 3).

The relatively low magnitude of association between SNPs in this gene-set and CTDs (i.e., odds ratio range: 0.7–1.5), suggests that the association between CTDs and the cytoskeletal gene-set observed in this study is unlikely to be driven by linkage disequilibrium between common and rare damaging variants with large effect sizes. Further, of the 1431 CTD cases included in our analyses, only 29 were included, and found to carry a rare damaging genotype in a cytoskeletal gene in the analyses of Watkins et al. [11] Hence, it is unlikely that the common-variant signal detected in the current analyses is driven by the same individuals that drove the rare-variant signal reported in Watkins et al. [11]

Post-hoc analyses were performed to assess whether the magnitude of the association between CTDs and the cytoskeletal gene-set was stronger when the set was restricted to include only those genes that were found to harbor rare damaging genotypes in cases with CHD in the analyses of Watkins et al. [11]. The genes included in each sub-set can be found in Appendix A. This table also includes the number of damaging de novo mutations and rare recessive or compound heterozygous genotypes in each gene, as reported by Watkins et al. [11] and the gene-level *p*-values generated in the MAGMA analysis of CTDs only.

Compared to the association between CTDs and the full cytoskeletal gene-set (β_S_ = 0.09, 95% CI 0.03–0.15), the magnitude of the association with the de novo subset (*N* = 82 genes, β_S_ = 0.12, 95% CI −0.05–0.30) was 1.3-fold higher (Table 4), and the magnitude of the association for the rare recessive or compound heterozygous subset was two-fold higher (*N* = 120 genes, β_S_ = 0.18, 95% CI 0.04–0.3). Further, when the cytoskeletal gene-set was restricted to include only those genes for which variants (de novo or recessive/compound heterozygous) were identified in more than one case (*N* = 50 genes: one gene with ≥2 de novo mutations only; 31 genes with ≥2 recessive or compound heterozygous genotypes only; 18 genes with at least one de novo mutation and one recessive or compound heterozygous genotype), the magnitude of the association with CTDs was 3.6-times higher (β_S_ = 0.32, 95% CI 0.08–0.56) than that for the full cytoskeletal gene-set. Similar results were obtained when this subset was further limited to include only the 39 genes for which recessive or compound heterozygous genotypes were observed in ≥2 cases (Table 4). Similar analyses conducted for the remaining 14 gene-sets revealed no clear patterns across datasets and none of the sub-set analyses had *p*-values less than our initial Bonferroni corrected value of *p* < 0.003.

The 10 genes with the lowest *p*-values in the subset of cytoskeletal genes that had rare variants in two or more cases are provided in Table 3. The genes in this subset include only one of the genes (Spire Type Actin Nucleation Factor 2, *SPIRE2*) with the 10 lowest *p*-values in the full cytoskeletal gene-set. The lowest gene association *p*-value in this sub-set was for *SPIRE2* (*p* = 0.012), which, along with *SPIRE1*, drives nucleation of actin filaments cells involved in intracellular vesicle transport [26]. Of the SNPs mapping to *SPIRE2*, the lowest *p*-value was *p* = 0.0004 for an intronic variant (rs12922448) with a scaled CADD score of 0.20. No variant in this gene had both a *p*-value < 0.05 and a CADD score ≥ 10.

## 4. Discussion

We assessed the associations of 15 CHD candidate gene-sets with CTDs and LVOTDs using summary statistics from analyses of common (i.e., MAF ≥ 5%) genetic variants. We found that, as a class, cytoskeletal genes were associated with CTDs. The cytoskeleton is involved in all aspects of cell shape changes and motility and is, therefore, critical for tissue morphogenesis and development. A role for cytoskeletal genes in the etiology of CHDs, including defects of the outflow tract, is supported by studies in animal models [27,28,29], and by the identification of potentially causal mutations in cytoskeletal genes in humans with CHDs [30,31]. In addition, among individuals with a CHD, the cytoskeletal gene-set has been found to be significantly enriched for damaging de novo mutations (*p* ≤ 7 × 10^−5^) as well as recessive or compound heterozygous (*p* ≤ 4 × 10^−5^) genotypes [11]. Hence, our study adds to the evidence for involvement of cytoskeletal genes in the etiology of CHDs in humans, and suggests that CHD risk may be influenced by common as well as rare variants in these genes.

Our study highlights the importance of conducting downstream analyses of genome-wide data that aggregate SNPs and genes into biologically meaningful groups and assess the overall effect of the set. In our data, no single SNP in the cytoskeletal set was strongly associated with CTDs (i.e., range of odds ratios: 0.7–1.5) and no SNP or gene in this set passed strict genome-wide, or even more lenient set-wide significance thresholds. Further, while the Z scores for genes in the cytoskeletal set were, on average, significantly higher (corresponding to lower *p*-values) than for all other genes, this difference was modest (i.e., β_S_ = 1.09). Hence, consistent with the observation that common genetic variants generally have only modest disease-associations, our analyses indicate that the association between CTDs and common variants in cytoskeletal genes is driven by weak signals across many genes, rather than by strong signals from a few genes.

Several cytoskeletal genes for which there is prior evidence of a role in heart development or an association with CHDs in humans had relatively low *p*-values in our analyses: *ACTA2* (*p* = 0.006), which is associated with bicuspid aortic valve in individuals with *ACTA2*-related thoracic aortic aneurisms [31] and with complex CHDs in individuals with *ACTA2*-related multisystemic smooth muscle dysfunction syndrome [32]; *NRP1* (*p* = 0.01), for which a homozygous splice site mutation was identified in a multiplex, consanguineous family with truncus arteriosus [33]; *RAC1* (*p* = 0.04), which is associated with a range of outflow tract defects in targeted knockout mice [27,28]; and *NOS3* (*p* = 0.09) which is associated with septal defects in the mouse knockout [34].

The relatively low *p*-values observed for cytoskeletal genes with prior links to CHDs provide some support for a causal interpretation of our observed association between CTDs and the cytoskeletal gene-set. Further support for such an interpretation is provided by the stronger association observed when our analyses were restricted to include only cytoskeletal genes with predicted damaging mutations in more than one CHD case, as compared to the full cytoskeletal gene set (i.e., β_S_ = 0.32 and β_S_ = 0.09, respectively), since the finding of deleterious genotypes in more than one affected individual increases the likelihood that a gene is truly disease-related.

In their rare variant burden analyses, Watkins et al. [11] observed enrichment of several gene-sets in addition to the cytoskeletal set—most notably, enrichment of de novo variants in the chromatin gene set (*p* ≤ 1 × 10^−5^) and enrichment of recessive and compound heterozygous genotypes in the cilia gene-set (*p* ≤ 1 × 10^−5^) [11]. In our analyses based on common variants, we found no evidence for association with the chromatin gene-set (CTDs, *p* = 0.15; LVOTDs, *p* = 0.63; CTDs + LVOTDs, *p* = 0.57) and only modest evidence for association with the cilia-gene set (CTD only, *p* = 0.03; LVOTD only, *p* = 0.09; CTD + LVOTD, *p* = 0.04). Watkins et al. also found some evidence of enrichment for de novo variants in the NOTCH signaling pathway gene-set (*p* ≤0.001) [11], which, while not formally significant after correction for multiple testing, had a relatively low *p*-value (*p* = 0.007) in our LVOTD only analyses. 

Based on their analyses of rare variants, Watkins et al. [11] concluded that different classes of genes contribute to CHDs via different mechanisms, with some gene-sets contributing predominantly via dominant (i.e., de novo) mutations (e.g., chromatin) and others via rare recessive or compound heterozygous genotypes (e.g., cilia). Our results extend these findings to suggest that some gene-sets also contribute to CHD risk via common variants. In the analyses of Watkins et al. [11], the cytoskeletal gene-set was one of only two sets (the other was *TGFβ* signaling) for which there was strong evidence of enrichment for both de novo mutations and rare recessive or compound heterozygous genotypes. As common variants are generally expected to have mild functional consequences, it may be that the developing heart is particularly vulnerable to variation in cytoskeletal genes.

In our analyses, the lack of association between CHDs and common variants in some gene-sets that were found to be enriched for rare variants (e.g., chromatin) may indicate that the impact of common variation on CHD risk is specific to particular gene-sets. However, the lack of association in this study could also be related to statistical power. In addition, the approach we used may underestimate true associations, since there may be a relatively large number of CHD-related genes outside of any given candidate gene-set. Moreover, each gene-set is likely to include a mixture of CHD relevant and non-relevant genes, which would also dilute the association signal. Further, although we considered the two largest categories of CHDs (i.e., CTDs and LVOTDs), it is possible that common variation in some gene-sets is associated with other CHD phenotypes. Such differences in the contribution of common variants across CHD phenotypes might also explain our observation of an association between the cytoskeletal gene-set and CTDs, but not LVOTDs.

## 5. Conclusions

Our analyses provide evidence for an association between CTDs and common variation within cytoskeletal genes. These findings highlight the importance of conducting downstream analyses of data from GWAS. Our findings also add to the evidence that cytoskeletal genes contribute to CHDs, and in particular to CTDs, in humans and suggest this gene-set may be somewhat unique in that variation across the spectrum, from rare to common, may contribute to risk. Given the evidence that CHD-related genetic variation includes a range of variant types (e.g., common, de novo, rare inherited), future studies aiming to enhance our understanding of the causes of CHDs should seek to capture all potentially relevant variation (e.g., common, de novo, rare inherited) as well as other potential etiologic complexities such as interactions within and between genes and gene-sets.

## Figures and Tables

**Table 1 genes-12-00655-t001:** CHD-related gene-sets analyzed for association with CTDs and LVOTDs.

Name	Description ^1^	# of Genes
Autism	High-ranking autism candidate genes	86
CHD	Non-cilia genes associated with congenital heart defects in humans or other organisms	402
Chromatin	Chromatin-modifying genes found to be disrupted in patients with congenital heart defects	163
Cilia	Expanded cilia gene list including the 302 SysCilia genes and potential cilia genes identified by a GOontology search in model organisms (zebrafish and mouse)	669
Cytoskeletal	Cytoskeleton genes identified using the Reactome pathway database, with exclusion of genes related to cilia structure or function	791
FGF signaling	Fibroblast growth factor signaling genes identified using the Reactome pathway database	87
FoxJ1	Genes with at least a two-fold change in expression when FoxJ1 is over-expressed or depleted in a zebrafish model	116
Hedgehog signaling	Hedgehog signaling genes identified using the Reactome pathway database	149
High heart expression	Genes with de novo mutations observed in human CHD cases and in the top quartile of expression in mouse embryonic day 14.5 hearts	146
Notch1	Hand curated Notch1 associated gene list	130
PDGF signaling	Platelet derived growth factor signaling genes identified using the Reactome pathway database	116
Ser-Thr kinases	Ser-Thr kinases identified using the Reactome pathway databases	47
Syscilia	Well-characterized structural cilia genes (SysCil 2.0) assembled from the literature	302
TGF-β	Assembled using the Reactome pathway database	431
WNT signaling	WNT signaling genes identified using the Reactome pathway databases	297

#, number. ^1^ Gene-sets as defined in Watkins et al. 2019 [11].

**Table 2 genes-12-00655-t002:** Summary of gene-set analyses for CTDs only (*N* = 1431 cases), LVOTDs only (*N* = 509 cases), and CTDs and LVOTDs (*N* = 1940) combined.

Gene-Set	# Genes Analyzed (# of Genes in Set) ^1^	CTDs Only (3 Datasets/1431 Cases) ^2^			LVOTDs Only (2 Datasets/509 Cases) ^2^			CTDs and LVOTDs (5 Datasets/1940 Cases) ^2^		
		Β_S_	95% CI	*p*-Value ^3^	β_S_	95% CI	*p*-Value ^3^	β_S_	95% CI	*p*-Value ^3^
Autism	76 (86)	−0.12	−0.03–0.06	0.89	0.11	−0.09–0.31	0.12	−0.02	−0.20–0.16	0.59
CHD	364 (402)	0.04	−0.05–0.11	0.21	0.05	−0.03–0.13	0.14	0.02	−0.06–0.10	0.36
Chromatin	148 (163)	0.07	−0.05–0.19	0.15	−0.02	−0.16–0.12	0.63	−0.01	−0.13–0.11	0.57
Cilia	612 (669)	0.06	0.001–0.12	0.03	0.04	−0.02–0.10	0.09	0.06	0.001–0.12	0.04
Cytoskeletal	726 (791)	0.09	0.03–0.15	0.001	−0.06	−0.12–0.001	0.97	0.04	−0.02–0.10	0.08
FGF signaling	83 (87)	0.03	−0.15–0.21	0.37	0.008	−0.17–0.18	0.46	−0.07	−0.23–0.09	0.80
FoxJ1	105 (116)	0.06	−0.08–0.20	0.20	0.06	−0.08–0.20	0.20	0.05	−0.09–0.19	0.24
Hedgehog signaling	137 (149)	0.11	−0.01–0.19	0.04	0.06	−0.06–0.18	0.19	0.05	−0.07–0.17	0.21
High heart expression	133 (146)	−0.12	−0.26–0.02	0.96	−0.03	−0.16–0.12	0.66	0.02	−0.12–0.16	0.41
Notch1	120 (130)	−0.05	−0.19–0.09	0.77	0.18	0.04–0.32	0.007	−0.12	−0.26–0.02	0.95
PDGF signaling	101 (116)	−0.11	−0.27–0.05	0.92	0.08	−0.08–0.24	0.15	−0.18	−0.34–0.02	0.99
Ser-Thr kinases	41 (47)	−0.06	−0.31–0.19	0.66	−0.03	−0.28–0.22	0.60	−0.13	−0.37–0.13	0.84
SysCilia	280 (302)	0.04	−0.06–0.14	0.19	0.04	−0.06–0.14	0.20	0.01	−0.07–0.09	0.40
TGF-β	402 (431)	−0.03	−0.11–0.05	0.77	−0.05	−0.13–0.03	0.88	−0.04	−0.12–0.04	0.86
WNT signaling	275 (297)	0.08	−0.02–0.18	0.04	−0.02	−0.12–0.08	0.65	0.05	−0.05–0.15	0.16

#, number; CI, confidence interval; CTDs, conotruncal heart defects; LVOTDs, left ventricular outflow tract defects. ^1^ # of genes in the set that were represented in our data (# of genes in set as specified in Watkins et al. 2019). ^2^ Number of datasets and total number of cases included in the meta-analyses that provided the summary statistics used as the initial input for these analyses. ^3^ Test of the null hypothesis that the mean association of the phenotype with the genes in the set is greater than that of genes not in the set (i.e., H_0_: β_S_ = 0 versus H_1_: β_S_ > 0).

**Table 3 genes-12-00655-t003:** Top 10 associations in the analyses of CTDs (*N* = 1431 cases) and all genes in the cytoskeleton gene-set and in the sub-set with rare, putatively damaging variants in > 2 cases ^1^ with a congenital heart defect.

Gene Symbol	Gene Name	*p*-Value	# De Novo ^1^	# Recessive or Compound Heterozygous ^1^	Total # of Rare Variants ^1^
Top 10 gene-associations in the full cystoskeletal gene-set (*N* = 726)					
CASS4	Cas scaffold protein family member 4	0.003	0	0	0
CLIP1	Cap-gly domain containing linker protein	0.006	0	0	0
ACTA2	Actin α 2, smooth muscle	0.006	0	0	0
KAZN	Kazin, periplankin interaction protein	0.007	0	0	0
MAEA	Macrophage erythroblast attacher, E3 ubiquitin ligase	0.010	0	0	0
TBC1D21	TBC1 domain family member 21	0.010	0	0	0
NRP1	Neuropilin 1	0.010	0	0	0
SPIRE2	Spire type actin nucleation factor 2	0.012	0	2	2
SEPT9	Septin 9	0.014	0	0	0
CLIC5	Chloride intracellular channel 5	0.014	0	0	0
Top 10 gene-associations in the sub-set of cystoskeletal genes with damaging rare genotypes in ≥ 2 cases ^1^ (*N* = 50)					
SPIRE2	Spire type actin nucleation factor 2	0.012	0	2	2
TNS1	Tensin 1	0.035	1	2	3
SCNN1D	Sodium channel epithelial 1 subunit delta	0.040	0	2	2
RAPH1	Ras association and pleckstrin homology domains 1	0.046	1	1	2
TENM2	Teneurin transmembrance protein 2	0.049	0	2	2
TACC2	Tranforming acidic coiled-coil containing protein 2	0.050	0	2	2
PLEC	Plectin	0.060	0	8	8
TRIP6	Thyroid hormone receptor interactor 6	0.073	0	2	2
NOS3	Nitric oxide synthase 3	0.086	0	2	2
BSN	Bassoon presynaptic cytomatrix protein	0.093	1	4	5

^1^ As reported in Watkins et al. [11]; includes cases with any type of CHD.

**Table 4 genes-12-00655-t004:** Summary of post hoc analyses of the association between CTDs and the cytoskeletal gene-set and sub-sets.

Gene-Set/Sub-Set	# of Genes	β_S_	95% Confidence Interval	*p*-Value ^2^
Full cytoskeletal gene-set	726	0.09	0.03–0.15	0.001
Subset with de novo mutations ^1^	82	0.12	−0.05–0.30	0.09
Subset with rare recessive mutations ^1^	120	0.18	0.04–0.32	0.007
Subset with more than one reported de novo or recessive mutations	50	0.32	0.08–0.56	0.002
Subset with more than one reported recessive mutation	39	0.32	0.08–0.56	0.005

#, number; ^1^ 18 genes are in both the de novo and recessive sub-sets. ^2^ Test of the null hypothesis that the mean association of the phenotype with the genes in the set is greater than that of genes that are not in the set (i.e., H_0_: β_S_ = 0 versus H_1_: β_S_ > 0).

## Data Availability

The genotype data used in these studies are available at: Pediatric Cardiac Genomics Consortium: https://www.ncbi.nlm.nih.gov/projects/gap/cgi-bin/study.cgi?study_id= phs001194.v2.p2. CHOP pediatric controls: https://www.ncbi.nlm.nih.gov/projects/gap/cgi-bin/study.cgi?study_id=phs000490.v1.p1. CHOP CTDs: https://www.ncbi.nlm.nih.gov/projects/gap/cgi-bin/study.cgi?study_id=phs000881.v1.p1. CHOP LVOTDs: https://www.ncbi.nlm.nih.gov/projects/gap/cgi-bin/study.cgi?study_id=phs000781.v1.p1 all web accessed on 27 April 2021.

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
