# Peer review of "Common Variation in Cytoskeletal Genes Is Associated with Conotruncal Heart Defects"

_genes, 2021, doi:10.3390/genes12050655_

Round 1

Reviewer 1 Report

Comments to the authors:

            I have embedded multiple comments in the PDF file that may help the authors further improve their work.  The PDF formatted version of the manuscript was incredibly hard to read:  numbered references were not noted as superscript; Greek symbols were left as blanks; Latin words were not italicized; nor were gene names (which is usual style).   

            I appreciate the challenge faced by these authors:  Despite their impressively large sample size standard genome-wide association tests failed to yield clear signals of potentially causal genes for CHD, perhaps due to the underlying complex and heterogeneous nature of this most-common birth defect.  Here they build upon previous work on this data set by switching to gene-set analysis of combined groups of the most common forms of CHD: conotruncal heart defects (1431 case-parent trios) and left ventricular outflow tract defects (509 case-parent trios) which represent 60% of all CHD cases.  I think their analytical strategy is sound although the description of the statistical methods does a poor job of fully explaining exactly what tests were used where.  They seem to be relying of names of programs to convey their meaning rather than defining the statistical models for the reader.

            While I think this is manuscript represents a competent analysis of a very valuable data resource, it is need of revision to maximize the benefit to the reader.

Author Response

Reviewer 1 Comments to the authors:

COMMENT: I have embedded multiple comments in the PDF file that may help the authors further improve their work.  The PDF formatted version of the manuscript was incredibly hard to read:  numbered references were not noted as superscript; Greek symbols were left as blanks; Latin words were not italicized; nor were gene names (which is usual style).   

RESPONSE: We regret that there were formatting issues with the PDF version of our manuscript. These issues appear to have arisen during the creation of the PDF, as they were not present in our original Word document. We have been in contact with Zanna Zhang in the Genes Editorial Office. She will review our revised Word and PDF documents to ensure the latter is accurate.

We have corrected errors (e.g. wrong tense, missing words) as identified by this reviewer. In addition, we have made several of the wording changes that this reviewer embedded in the PDF.  In particular, we made changes when we agreed that the suggested revisions improved the clarity of the text and opted not to make changes where we felt the suggested revisions either changed the meaning or did not improve the clarity of the text. We will, however, defer to the editorial staff on specific wording choices. 

We have responded to each of the embedded comments within the attached PDF named: genes_1181193_review1_response.

COMMENT: I appreciate the challenge faced by these authors:  Despite their impressively large sample size standard genome-wide association tests failed to yield clear signals of potentially causal genes for CHD, perhaps due to the underlying complex and heterogeneous nature of this most-common birth defect.  Here they build upon previous work on this data set by switching to gene-set analysis of combined groups of the most common forms of CHD: conotruncal heart defects (1431 case-parent trios) and left ventricular outflow tract defects (509 case-parent trios) which represent 60% of all CHD cases.  I think their analytical strategy is sound although the description of the statistical methods does a poor job of fully explaining exactly what tests were used where.  They seem to be relying of names of programs to convey their meaning rather than defining the statistical models for the reader.

RESPONSE: We reviewed and believe that we have provided the relevant details for the gene and gene-set analyses conducted as part of this study (Methods, Gene-set Analyses). However, we agree that details were lacking for the Methods used to generate the SNP-level summary statistics (reported in Agopian et al. 2018) that were used as input for the gene and gene-set analyses described in this paper.  Consequently, we have added details to the description of the SNP-level analyses that were used to generate the input values for these studies (See: Methods, Data sets, paragraph 4). Further details of the SNP-level analyses are provided in our prior publication (Agopian et al. 2018), which is also referenced in this section of the manuscript.

Reviewer 2 Report

Musfee et al examine the role of common genetic variation in congenital heart disease. While rare and de novo mutation undoubtedly cause CHD, the evidence for common variation is poor. GWAS have generally not been very informative, which is not surprising given that deleterious variation of large effect undoubtedly face strong negative selection. To detect small effects, the authors evaluate common variation in sets of genes that share common functions or pathways, hypothesizing that the aggregate of small effects in a gene set is detectable. They test the hypothesis in >2000 patients who have conotruncal heart defects or left ventricular obstructive lesions. Their analysis is novel. The authors discover a significant association of common genetic variation in cytoskeletal genes in conotruncal heart defects. While a common variant by intself is unlikely to cause a heart defects (the authors do claim that they do), the results are of interest and relevant to a long presumed, but inadequately investigated complex genetic basis of congenital heart disease. The two questions that I raise below are meant to spur further interest in the topic.

  1. Negative results are obtained all the other gene sets examined despite clear evidence for the role of rare mutation in some of them, e.g., chromatin, cilia, TGF signaling, as presented in Watkins 2019. The results could be misleadingly negative, however, if common variation in only a subset of the genes in a set is relevant. Just as the authors demonstrated stronger association when the cytoskeletal gene set was restricted previously implicated genes, it would be worthwhile to know whether similar analyses would likewise strengthen an association in the negative gene sets.
  2. Would the authors care to discuss where their work can go in the future or alternative methods to evaluate the role of common genetic variation in CHD? Specifically, I note that a common variant with small effect is unlikely to cause CHD by itself. Pathological interactions between common variants, if they play any role at all, and other genes are more likely. Without a vision for the path forward, the field will be stuck on rare variants and large effect sizes.
